# Fabrication of Noble-Metal-Free Mo_2_C/CdIn_2_S_4_ Heterojunction Composites with Elevated Carrier Separation for Photocatalytic Hydrogen Production

**DOI:** 10.3390/molecules28062508

**Published:** 2023-03-09

**Authors:** Hong Qiu, Xiaohui Ma, Hongxia Fan, Yueyan Fan, Yajie Li, Hualei Zhou, Wenjun Li

**Affiliations:** Beijing Key Laboratory for Science and Application of Functional Molecular and Crystalline Materials, University of Science and Technology Beijing, Beijing 100083, China

**Keywords:** noble-metal-free, Mo_2_C, CdIn_2_S_4_, heterojunction, carrier separation

## Abstract

Molybdenum-based cocatalyst being used to construct heterojunctions for efficient photocatalytic H_2_ production is a promising research hotspot. In this work, CdIn_2_S_4_ was successfully closely supported on bulk Mo_2_C via the hydrothermal method. Based on their matching band structures, they formed a Type Ⅰ heterojunction after the combination of Mo_2_C (1.1 eV, −0.27 V, 0.83 V) and CdIn_2_S_4_ (2.3 eV, −0.74 V, 1.56 V). A series of characterizations proved that the heterojunction composite had higher charge separation efficiency compared to a single compound. Meanwhile, Mo_2_C in heterojunction could act as an active site for hydrogen production. The photocatalytic H_2_ production activity of the heterojunction composites was significantly improved, and the maximum activity was up to 1178.32 μmmol h^−1^ g^−1^ for 5Mo_2_C/CdIn_2_S_4_ composites. 5Mo_2_C/CdIn_2_S_4_ heterojunction composites possess excellent durability in three cycles (loss of 6%). Additionally, the mechanism of increased activity for composites was also investigated. This study provides a guide to designing noble-metal-free photocatalyst for highly efficient photocatalytic H_2_ evolution.

## 1. Introduction

With the increasingly serious energy crisis, H_2_ energy is attracting more and more attention due to its renewable, clean, high energy density, and so on [1,2,3]. As a feasible method, photocatalytic H_2_ production from water has been a research hotspot among all the methods of H_2_ production [4,5,6]. Proverbially, developing stable and high-efficiency visible light photocatalyst is always a core challenge for photocatalysis technology [7,8]. To date, numerous semiconductors—including sulfides, metal oxides, and nitrides—have been extensively exploited for photocatalytic H_2_ production [9,10,11]. Among the developed photocatalysts, CdIn_2_S_4_ (CIS) is of great interest [12,13,14]. Nevertheless, for bare CIS, the low carrier separation efficiency and lack of active site still need to be urgently solved. To address those issues, various methods have been carried out to obtain elevated photocatalytic activity of CdIn_2_S_4_-based photocatalysts, such as tuning the morphologies [15], constructing heterojunctions [16,17], doping metal or nonmetal elements [18,19], and so on. He et al. [20] prepared ultra-thin CdIn_2_S_4_ nanosheets to acquire efficient photocatalytic activity. Chen et al. [21] designed CdIn_2_S_4_/TiO_2_ Z-scheme heterojunction with high carrier separation efficiency for photocatalytic H_2_ production. Yu et al. [17] reported that PdS-loaded ZnIn_2_S_4_/CdIn_2_S_4_ flower-like microspheres had significantly improved photocatalytic activity and high-efficiency stability under aqueous Na_2_SO_3_ and Na_2_S solution. Guo et al. [12] prepared CdIn_2_S_4_/CNFs/Co_4_S_3_ nanofiber networks with efficient charge separation and adequate active sites for H_2_ production using solar-driven water splitting. Additionally, cocatalysts are critical for booting carrier separation efficiency and causing the active site of H_2_ production to increase the activity of photocatalysts. Proverbially, noble metals (e.g., Pt, Au) have been proven to be the most effective cocatalysts for photocatalytic H_2_ production [22]. Regrettably, the scarcity of noble metals greatly restricts their extensive utilization. Accordingly, exploiting accessible and low-priced cocatalysts is of great significance for achieving high-efficiency photocatalytic H_2_ production.

As a type of transition metal carbide, Mo_2_C has been widely studied in the field of electrocatalysis due to its good electric conductivities and high catalytic properties [23,24]. Universally, electrocatalysts with superior activity can also be used as efficient cocatalysts for photocatalytic hydrogen production. Therefore, Mo_2_C has great potential for photocatalytic H_2_ evolution [25,26,27]. In our previous study, our group prepared In_2_S_3_@Mo_2_C heterojunction for photocatalytic H_2_ generation, revealing that Mo_2_C was a highly H_2_-generation-active cocatalyst [27]. Yue et al. [28] designed a SrTiO_3_@Mo_2_C core-shell nanostructure with enhanced charge separation for dramatically elevated photocatalytic H_2_ evolution. Yue et al. [29] prepared dandelion-like Mo_2_C/TiO_2_ heterojunction photocatalysts with efficient charge separation and catalytically active sites for photocatalytic H_2_ evolution. Zhang et al. [30] reported g-C_3_N_4_/Mo_2_C hybrid photocatalysts in which, as a cocatalyst, Mo_2_C on the surface of g-C_3_N_4_ leads to promoted charge separation, improved visible light absorption, and enhanced following H_2_-evolution rate. In this sense, after Mo_2_C is combined with matched CdIn_2_S_4_, it may be an ideal heterojunction system for enhanced photocatalytic H_2_ generation. In heterojunction, Mo_2_C could not only accelerate the separation efficiency of photogenerated carriers but could also act as an active site of H_2_ production. Moreover, it is worth noting that the Mo_2_C/CdIn_2_S_4_ system has not been reported for photocatalytic performance.

In this work, we successfully synthesized a novel non-noble-metal Mo_2_C/CdIn_2_S_4_ heterostructure for photocatalytic H_2_ production in which CdIn_2_S_4_ nanoparticles were loaded on bulk Mo_2_C. Under solar light irradiation, in heterojunction, Mo_2_C could trap photo-generated electrons of CdIn_2_S_4_ to efficiently accelerate carrier separation. Meanwhile, the electrons could reduce water to hydrogen at the active site of Mo_2_C. As a result, the Mo_2_C/CdIn_2_S_4_ composites displayed visibly elevated H_2_ production activity up to 1178.32 μmmol h^−1^ g^−1^. Highly significantly, the designed heterojunction composites possessed outstanding stability, and the mechanism of photocatalytic activity enhancement was investigated. This work offers a thinking to design non-noble-metal heterostructure with high carriers separation efficiency for photocatalytic H_2_ production.

## 2. Results and Discussion

The morphologies of all samples were obtained via scanning electron microscopy (SEM), transmission electron microscopy (TEM), and high-resolution transmission electron microscopy (HRTEM). Figure 1A showed that CdIn_2_S_4_ had nanoparticle structures with sizes of 25–180 nm, and Figure 1B displayed that Mo_2_C had a bulk structure about 1–2 μm in size. Figure 2A,B also showed that CdIn_2_S_4_ and Mo_2_C had nanoparticle structures and bulk structures, which confirmed the results of scanning electron microscopy. Figure 1C clearly revealed that CdIn_2_S_4_ was deposited on the surface of Mo_2_C. Simultaneously, compared to the pure molybdenum carbide, Figure 2C shows that the massive molybdenum carbide edge had many CdIn_2_S_4_ particles, which also confirmed the results of the scanning electron microscopy. The results of high-resolution transmission electron microscopy were analyzed to further confirm the formation of MS heterojunction composites. Figure 2D,E represented that the lattice fringe spacings of (3 1 1) planes (CdIn_2_S_4_) and (1 0 1) planes (Mo_2_C) were 0.33 nm and 0.21 nm, respectively. More significantly, Figure 2F possessed two lattice fringes of 0.33 nm and 0.21 nm assigned to the (3 1 1) planes of CdIn_2_S_4_ [31] and the (1 0 1) planes of Mo_2_C [27], respectively, which demonstrated the simultaneous presence of CdIn_2_S_4_ and Mo_2_C in the MS heterojunction composites. In a word, the aforementioned results corroborated the successful synthesis of the heterojunction composites.

In Figure 3, the optical properties of CdIn_2_S_4_, Mo_2_C, and the 5MS hybrid were discreetly studied via UV–Vis diffuse reflectance spectra. As expected, CdIn_2_S_4_ possessed an evident edge at about 575 nm that was derived from its transition of the band [15]. Bare Mo_2_C was black and revealed a broad and strong visible light absorption capacity from 300 nm to 800 nm. Interestingly, it was found that the absorption edge of heterojunction composites extended to longer wavelength regions compared to bare CdIn_2_S_4_. Meanwhile, the visible light absorption intensity of heterojunction composites was visibly boosted at 500–800 nm with increasing Mo_2_C content. The stronger visible light absorption capacity plays a pivotal role in achieving solar energy conversion. Based on our previous research [27], the band gaps (E_g_) of Mo_2_C and CdIn_2_S_4_ were 1.1 eV and 2.3 eV, respectively, which conform to the literature values well [29,31]. It can be clearly seen that the band gap of MS heterojunction composites is reduced compared to CdIn_2_S_4_ in Appendix A. As shown in Figure 3B, at a frequency of 1 kHz, Mott–Schottky (M-S) tests were carried out to analyze the type of semiconductors and the charge transfer process. The straight lines of Mo_2_C and CdIn_2_S_4_ both have positive slopes, which suggests that they are intrinsic n-type characteristics. Additionally, the conduction band potential values are confirmed on the basis of extrapolation of the straight line toward the x-axis. As everyone knows, the flat band potential is around equal to the conduction band potential for semiconductors with intrinsic n-type characteristics. In line with the intercept of the M-S plots (Figure 3B,C), the conduction band potentials (E_CB_) of Mo_2_C and CdIn_2_S_4_ are and −0.27 V and −0.74 V, respectively. Obviously, the conduction band potential of Mo_2_C is more positive than the conduction band potential of CdIn_2_S_4_, and this difference in energy level will help drive electron transfer from the conduction band potential of CdIn_2_S_4_ to the conduction band potential of Mo_2_C for photocatalytic hydrogen production. Therefore, on the basis of the following formula, the valence bands (E_VB_) of Mo_2_C and CdIn_2_S_4_ are 0.83 V and 1.56 V, respectively.
(1)$$E_{VB}=E_{CB}+E_{g}$$

The phase of prepared samples is authenticated and discussed via X-ray diffraction patterns (XRD), and the results are illustrated in Figure 4. All as-prepared samples possessed sharp diffraction peaks, indicating excellent crystallinity. The patterns of CdIn_2_S_4_ and Mo_2_C were confirmed to be cubic (JCPDF: 27-0060) and hexagonal (JCPDF: 35-0787) structures, respectively [26,32]. The peaks around 2θ of 14.2°, 23.1°, 27.3°, 33.1°, 43.3°, and 47.4° were readily instructed to those of CdIn_2_S_4_ and accorded to the (1 1 1), (2 2 0), (3 1 1), (4 0 0), (4 4 0), and (6 2 2) planes, respectively [32]. For bare Mo_2_C, the peaks at 34.4°, 38.0°, 39.4°, 52.2°, 61.5°, and 69.5° were observed and assigned to the (1 0 0), (1 0 1), (1 1 0), (0 0 2), (2 0 0), and (2 0 1) planes, respectively [26]. The heterojunction composites still held the evident characteristic peaks of CdIn_2_S_4_ ((2 2 0), (3 1 1), (4 0 0), (4 4 0), and (6 2 2) planes) after the hybridization of CdIn_2_S_4_ and Mo_2_C. Although the peaks of Mo_2_C have low contents, the (1 0 0), (1 0 1), (1 1 0), and (0 0 2) planes of Mo_2_C could be observed at 34.4°, 38.0°, 39.4°, and 52.2°, respectively in heterojunction composites [27]. The peak intensity of Mo_2_C significantly increased along with the elevation of the Mo_2_C content, which was advantageous for photocatalytic hydrogen production and had been studied in previous reports. The X-ray diffraction patterns (XRD) results demonstrated that MS composites were successfully fabricated, which is consistent with scanning electron microscopy (SEM), transmission electron microscopy (TEM), and high-resolution transmission electron microscopy (HRTEM) results.

Subsequently, we also utilized X-ray photoelectron spectroscopy (XPS) to further analyze and study the composition of the photocatalysts as well as the strong electronic interactions between CdIn_2_S_4_ and Mo_2_C. Figure 5A displays the full XPS survey of 5MS composites, and the full XPS survey explains the existence of Mo, Cd, C, S, and In elements in the ternary composites, which shows that the heterojunction composites are prepared successfully. In Figure 5B, for bare CdIn_2_S_4_, the peaks at 412.2 eV and 405.5 eV correspond to Cd 3d_3/2_ and Cd 3d_5/2_, respectively, and In 3d_3/2_ and In 3d_5/2_, respectively, which are the characteristic peaks of Cd^2+^ and In^3+^ in CdIn_2_S_4_. In Figure 5C, for bare CdIn_2_S_4_, the peaks at 452.5 eV and 444.9 eV correspond to In 3d_3/2_ and In 3d_5/2_, respectively, which are the characteristic peaks of In^3+^ in CdIn_2_S_4_. In Figure 5B, for 5MS composites, the peaks at 412.1 eV and 405.3 eV correspond to Cd 3d_3/2_ and Cd 3d_5/2_, respectively, which are the characteristic peaks of Cd^2+^ in the ternary composites. In Figure 5C, for 5MS composites, the peaks at 452.3 eV and 444.8 eV correspond to In 3d_3/2_ and In 3d_5/2_, respectively, which are the characteristic peaks of In^3+^ in the ternary composites. In Figure 5D, the S 2p data of CdIn_2_S_4_ centered at binding energies of 162.9 eV and 161.7 eV match with S 2p_1/2_ and S 2p_3/2_, respectively. In Figure 5D, the S 2p data of 5MS heterojunction composites centered at binding energies of 162.8 eV and 161.6 eV match with S 2p_1/2_ and S 2p_3/2_, respectively. Nevertheless, a distinct peak at about 169.0 eV could be seen, distributing SO_4_^2−^ resulting from hydrothermal processes in Figure 5D. For Mo 3d spectra of bare Mo_2_C (Figure 5E), there were three peaks of Mo 3d in Mo_2_C situate at 233.0 eV, 228.7 eV, and 236.1 eV, corresponding to Mo 3d_3/2_, Mo 3d_5/2_, and Mo-O band, respectively. It can also be seen that Mo mainly exists in three forms (Mo 3d_5/2_ and 227.7 eV, Mo 3d_3/2_ and 232.5 eV, Mo-O and 235.7 eV) in 5MS composites in Figure 5E. Furthermore, in Figure 5E, there is a distinct peak of 225.9 eV in 5MS composites, which can be attributed to S 2s, which is also consistent with our previous research and what others have reported. In Figure 5F, for C 1s spectra of Mo_2_C, in addition to the carbon standard peak (284.8 eV), there are two peaks located at 286.4 eV and 288.9 eV, assigned to C-O and C=O, respectively. The binding energies of C-O and C=O in 5MS composites are 286.8 eV and 288.9 eV, respectively, in Figure 5F. After the hybridization of CdIn_2_S_4_ and Mo_2_C, the binding energies of all elements have shifted slightly compared to pure CdIn_2_S_4_ and Mo_2_C, indicating a strong interaction between CdIn_2_S_4_ and Mo_2_C, which is very advantageous for photocatalytic hydrogen production [33,34]. The X-ray photoelectron spectroscopy (XPS) results support the X-ray diffraction patterns (XRD), scanning electron microscopy (SEM), transmission electron microscopy (TEM), and high-resolution transmission electron microscopy (HRTEM) results.

The photocatalytic behavior of all samples was assessed via photocatalytic experiment under visible light irradiation (λ ≥ 420 nm) (Figure 6). CdIn_2_S_4_ revealed low photocatalytic activity due to the scarce active site and rapid carrier recombination. Additionally, for Mo_2_C, there was no photocatalytic activity because of the rapid carrier recombination resulting from a narrow band gap, coinciding well with the results reported [35]. Extraordinarily, after CdIn_2_S_4_ was incorporated with Mo_2_C, the photocatalytic activities of MS composites increased sharply. The above H_2_ production process also confirms the results of photocatalysis and electrochemistry, that is, after the combination of Mo_2_C and CdIn_2_S_4_, not only was the photogenerated carrier separation efficiency improved, but the H_2_ production activity was surprisingly increased as well. In Figure 6A, the amount of H_2_ produced was severally 973.4 μmol/g (1MS), 3681.5 μmol/g (3MS), 6313.6 μmol/g (5MS), and 4596.9 (7MS) μmol/g in 5 h. In Figure 6B, the corresponding average H_2_ production rate of MS composites was 181.67 μmol h^−1^ g^−1^, 687.08 μmol h^−1^ g^−1^, 1178.32 μmol h^−1^ g^−1^, and 857.94 μmol h^−1^ g^−1^, respectively. Notably, when the content of Mo_2_C increased, the H_2_ production rates displayed a volcano-shaped photoactivity trend. The best weight ratio of Mo_2_C was determined to be 5 wt%, and the photocatalytic activity could reach the maximum value. This obvious improvement might stem from (i) Mo_2_C acting as active sites for water reduction; (ii) the separation efficiency of MS heterojunction structures being distinctly boosted. However, after the content of Mo_2_C exceeds 5%, the photocatalytic activity gradually reduced. This is because high Mo_2_C can not only block visible light absorption of CdIn_2_S_4_ but can also serve as the recombination center of carriers, resulting in the suppression of electron–holes separation [30,36]. The above results imply the appropriate Mo_2_C is significant for optimizing the photocatalytic activity of photocatalyst. Moreover, as shown in Figure 6B, the stability tests of photocatalytic hydrogen production for 5MS composites are also determined. There was a slight drop in hydrogen production per cycle, probably due to the loss of the photocatalyst during filtration and washing. Significantly, in three photocatalytic tests, 5MS composites reveal highly stable photocatalytic performance (Figure 6C). The comparison of photocatalytic performance between this work and the current work with similar work was in Appendix A.

To explore the carrier separation of photocatalysts, the PL and a series of electrochemical tests are obtained and studied (Figure 7). Generally, photoluminescence (PL) spectroscopy is one of the commonly used methods for evaluating the efficiency of photo-generated electron–hole pair charge separation. Due to its intrinsic properties, different photocatalysts have different emission spectra. For photocatalysts, the lower PL emission peak intensity means the lower recombination of photogenerated charge [37]. Figure 7A display that MS composites possess lower peak than bare CdIn_2_S_4_, which means that composites have higher carrier separating efficiency after the CdIn_2_S_4_ is supported on the surface of Mo_2_C cocatalyst. Therefore, the addition of Mo_2_C can increase the carrier separation of the catalyst. Figure 7B reveals the transient photocurrent responses (TPR) of all photocatalysts to investigate the charge transfer under simulated solar irradiation. Under visible light, the photocurrent responses of all photocatalysts were improved. As expected, it can be seen that all MS composites have stronger photocurrent than CdIn_2_S_4_, effectively implying charge transfer for composites [38]. The order of photocurrent intensity is: CdIn_2_S_4_ < 1MS < 3MS < 7MS < 5MS. The orders are in keeping with the photocatalytic H_2_ production results. The higher photocurrent indicates faster and more efficient e^−^ migration from CB of CdIn_2_S_4_ to CB of Mo_2_C, followed by a reduction reaction. In addition, the photocurrent result is in good agreement with results of PL above and EIS below. The electrochemical impedance spectroscopy (EIS) was measured and recorded on the open circuit potential under visible light. The EIS Nyquist plot is also used to understand the charge transfer of photocatalysts [39]. As shown in Figure 7C, compared to pure CdIn_2_S_4_, the semicircle curves of EIS for the MS heterojunction composites are shown, which indicates that photoinduced carriers are tardy recombination and affect migration with the addition of Mo_2_C. The order of radius is: 5MS < 7MS < 3MS < 1MS < CdIn_2_S_4_, matching with the photocurrent results. The illustration in Figure 7C is a model circuit. Moreover, 5MS composites have the smallest arcs compared to other samples. This result implied 5MS composites’ presence faster than electron migration and interface electron migration resistance. In summary, the PL and electrochemical results of all samples illustrate that Mo_2_C could speed up charge separation efficiency to obtain many more free electrons for photocatalytic hydrogen production.

Based on band arrangement and characterization results, the probable photocatalytic mechanism of heterojunction composites is designed and explained to verify the elementary reason for the distinct amendment of the photocatalytic performance of the MS heterojunction photocatalyst (Figure 8). The low H_2_ activity of CdIn_2_S_4_ may be due to the scarce active site and rapid carrier recombination. In the MS heterojunction, Mo_2_C is an electron capture trap, which can quickly extract photoinduced electrons generated in CdIn_2_S_4_, accelerate the separation of photoinduced electron–hole pairs and reach higher photocatalytic hydrogen production. Clearly, a Type Ⅰ heterojunction could be formed because the conduction band of Mo_2_C is more positive than that of CdIn_2_S_4_ and the valence band of Mo_2_C is more negative than that of CdIn_2_S_4_. The electron of the valence band for Mo_2_C and CdIn_2_S_4_ could be stimulated to the conduction band by visible light. Due to excellent electrical conductivity, Mo_2_C can rapidly capture the electrons of CdIn_2_S_4_ before the charge recombination, so it appears that the electrons of conduction band for CdIn_2_S_4_ transferred to the conduction band of Mo_2_C. Electrons on Mo_2_C could reduce water to hydrogen (H^+^ + e^−^ → 0.5H_2_). In the public eye, the separation and recombination of charge carrier are two competitive processes. Considering the different transfer rates of e^−^ and h^+^ from CdIn_2_S_4_ to Mo_2_C and the excellent metallic conductivity of Mo_2_C, the h^+^ in the VB of CdIn_2_S_4_ partially transfers to the valence band of Mo_2_C, and the remaining h^+^ in the valence band of CdIn_2_S_4_ and the h^+^ in the valence band of Mo_2_C reacts with lactic acid, thereby reducing the surface charge recombination. The formation of a Type Ⅰ heterojunction greatly improves the separation rate of carriers, demonstrated clearly by PL and electrochemical results. This obvious improvement might stem from (i) Mo_2_C acting as active sites for water reduction and (ii) the separation efficiency of MS heterojunction structures being distinctly boosted. Based on these results, synergetic modification of carrier separation and active sites result in the remarkably elevated photocatalytic activity of MS heterojunction composites.

## 3. Experiment

### 3.1. Reagents

Molybdenum carbide (Mo_2_C, 325 mesh, ≥ 99.5%) was acquired from Saen Chemistry Technology (Shanghai, China). Indium(III) nitrate hydrate (In(NO_3_)_3_·3H_2_O), thioacetamide (TAA), cadmium nitrate (Cd(NO_3_)_2_·4H_2_O), and lactic acid were all obtained form from Aladdin.

### 3.2. Synthesis of CdIn_2_S_4_ and Mo_2_C-CdIn_2_S_4_

Synthesis of CdIn_2_S_4:_ Amounts of 2 mmol In(NO_3_)_3_·3H_2_O, 4 mmol thioacetamide, and 1 mmol Cd(NO_3_)_2_·4H_2_O were sequentially added to 70 mL deionized water. After stirring for 1 h, turbid liquids were transferred to a Teflon-lined steel autoclave. The resulting reaction was heated at 180 °C for 24 h. After cooling to room temperature, the products were collected via centrifugation, washed with deionized water and ethanol several times, and then dried at 60 °C for 12 h.

Synthesis of Mo_2_C-CdIn_2_S_4:_ Amounts of 2 mmol In(NO_3_)_3_·3H_2_O, 0.1175 g Mo_2_C, 4 mmol thioacetamide, and 1 mmol Cd(NO_3_)_2_·4H_2_O were sequentially added in 70 mL deionized water. Then, the turbid liquids were kept at 180 °C for 24 h. After cooling to room temperature, the products were collected via centrifugation, washed with deionized water and ethanol several times, and then dried at 60 °C for 12 h. The heterojunction composites containing 0.2 g of CdIn_2_S_4_ with 0.01 g of Mo_2_C were labeled as 5MS photocatalysts. The other Mo_2_C/CdIn_2_S_4_ photocatalysts were prepared by introducing different amounts of Mo_2_C into the solution with other reaction parameters fixed. Similarly, the samples of CdIn_2_S_4_ with 1, 3, and 7 mass percent Mo_2_C are labeled as 1MS, 3MS, and 7MS, respectively. The additional experiments for optimization of the amounts and conditions were not used in the synthesis process.

### 3.3. Characterization

The crystal structures of pristine Mo_2_C, CdIn_2_S_4_, and Mo_2_C/CdIn_2_S_4_ were tested using Cu Ka radiation (I = 1.5406 Å, 40 kV and 40 mA) in X-ray diffraction patterns (XRD, D/Max-RB, Rigaku, Japan). Transmission electron microscopy (TEM), high-resolution transmission electron microscopy (HRTEM) (F-20, FEI, Hillsboro, OR, USA), and scanning electron microscopy (SEM) (S-4800; Hitachi, Tokyo, Japan) images were obtained. UV–Vis diffuse reflectance spectra (DRS) were carried out using a T9S spectrophotometer with BaSO_4_ as a reflectance standard. X-ray photoelectron spectroscopy (XPS) analysis was examined with a monochromatic X-ray source manufactured on an X-ray photoelectron spectrometer (Thermo Fisher Scientific K-Alpha, Waltham, MA, USA). The photoluminescence (PL) was observed using a fluorescence spectrophotometer (F-4500, Hitachi, Tokyo, Japan) with a Xe lamp as the excitation light source.

### 3.4. Electrochemical Measurements

The electrochemical impedance spectroscopy (EIS)–photocurrent measurement (TPR) curves of photocatalysts were obtained using a CHI660E electrochemical workstation. The samples, a saturated calomel electrode (SCE), and a Pt wire were employed as the working electrode, reference electrode, and counter electrode, respectively. All electrochemical tests used incandescent lamps under visible light. The aqueous solution of Na_2_SO_4_ (0.5 mol L^−1^) served as the electrolyte. In total, 5 mg of photocatalyst was suspended in mixed solution with ethanol and Nafion. The as-prepared samples were dispersed into a circle with a diameter of 6 mm on the bottom-middle of an ITO glass matrix using a micropipette and dried at room temperature.

### 3.5. Photocatalytic H_2_ Evolution Test

The photocatalytic H_2_ evolution reaction was operated in a typical reaction system according to our previous reports [40,41,42]. The photocatalytic experiment was carried out under visible light (λ ≥ 420 nm). The information of used Xenon lamp and the intensity of the incident radiation entering the photoreactor were added in Appendix A. First, 30 mL of lactic acid/H_2_O solution (10% (*v*/*v*) pH = 2.2) was prepared, then 30 mg samples were added in the above solution by ultrasound treatment for 30 min. Before irradiation, the air in reaction container was driven out through the high-purity argon (Ar) gas for 0.5 h. The amount of hydrogen generation was tested via gas chromatography (GC-7920, TCD). Furthermore, stability tests were also carried out to evaluate the stability of MS photocatalysts in three continuous experiments. For the next test, the samples were collected via centrifugation and washing thoroughly with ethanol and water several times and were then dried at 60 °C after every test for H_2_ generation.

## 4. Conclusions

In this research, the novel noble-metal-free Mo_2_C/CdIn_2_S_4_ composites were firstly developed via a facile fabrication process for highly efficient photocatalytic H_2_ production. The structure, morphology, and performance of heterojunction composites were analyzed using different characterization techniques. The visible light photocatalytic properties of the Mo_2_C/CdIn_2_S_4_ composites were investigated, and the best photocatalytic activity was up to 1178.32 μmmol h^−1^ g^−1^ for 5Mo_2_C/CdIn_2_S_4_ composites. Notably, the heterojunction composites possessed high stability in three cycles. The remarkable elevated photocatalytic activity may be due to the accelerated separation efficiency and more active sites for photocatalytic water splitting. We believe that Mo_2_C, as noble-metal-free cocatalyst to modify other photocatalysts, has great potential for efficient photocatalytic hydrogen production.

## Figures and Tables

**Figure 1 molecules-28-02508-f001:**
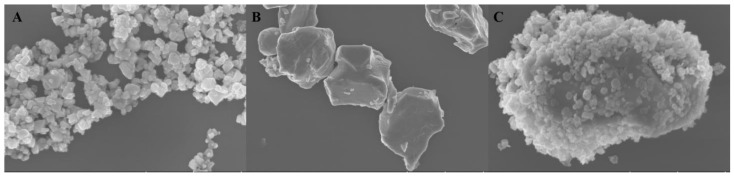
The scanning electron microscopy (SEM) of CdIn_2_S_4_ (**A**), Mo_2_C (**B**), and 5MS (**C**) heterojunction composites.

**Figure 2 molecules-28-02508-f002:**
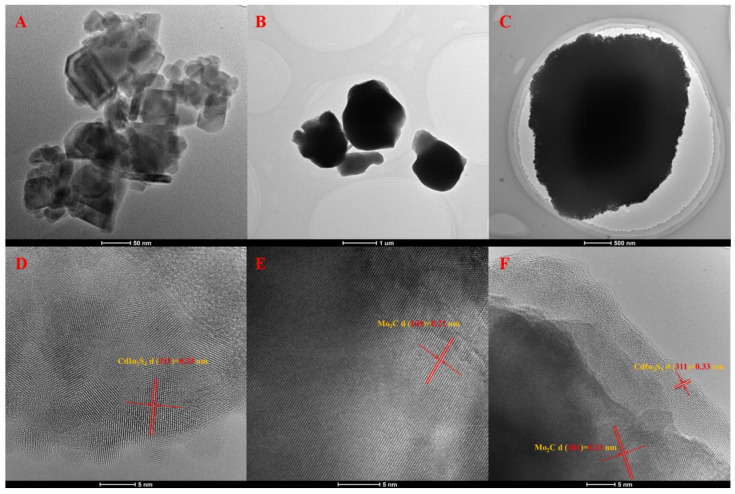
The transmission electron microscopy (TEM) and high-resolution transmission electron microscopy (HRTEM) of CdIn_2_S_4_ (**A**,**D**), Mo_2_C (**B**,**E**), and 5MS (**C**,**F**) heterojunction composites.

**Figure 3 molecules-28-02508-f003:**
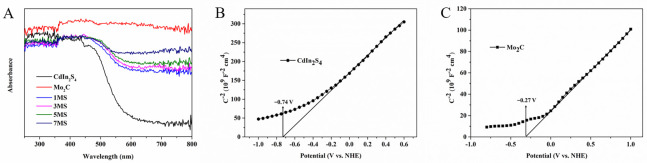
The UV−Vis diffuse reflectance spectra of CdIn_2_S_4_, Mo_2_C, and MS heterojunction composites (**A**); Mott−Schottky plots for CdIn_2_S_4_ (**B**) and Mo_2_C (**C**).

**Figure 4 molecules-28-02508-f004:**
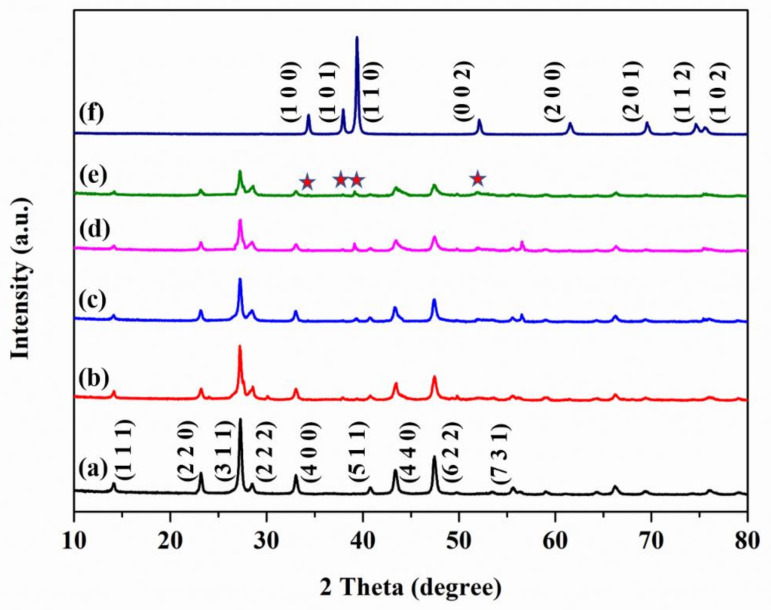
The X-ray diffraction patterns, of CdIn_2_S_4_, Mo_2_C, and MS heterojunction composites ((**a**) CdIn_2_S_4_; (**b**) 1MS; (**c**) 3MS; (**d**) 5MS; (**e**) 7MS; (**f**) Mo_2_C. 
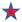
 is peaks of Mo_2_C, the (1 1 1), (2 2 0), (3 1 1), (4 0 0), (5 1 1), (4 4 0), (6 2 2), and (7 3 1) planes corresponds to CdIn_2_S_4_, the (1 0 0), (1 0 1), (1 1 0), (0 0 2), (2 0 0), (2 0 1), (1 1 2), and (1 0 2) planes corresponds to Mo_2_C).

**Figure 5 molecules-28-02508-f005:**
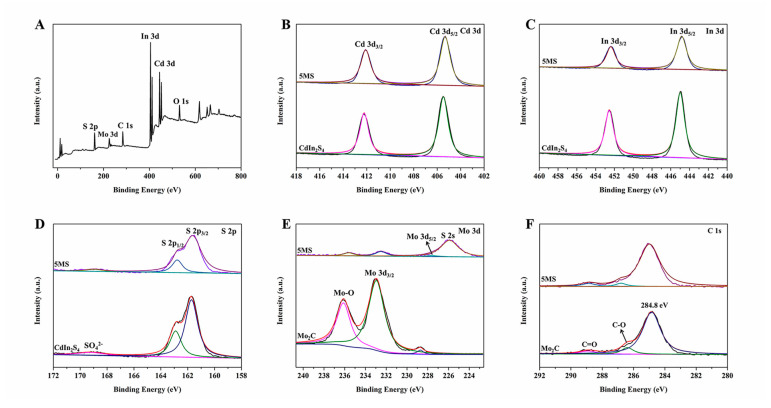
The X-ray photoelectron spectroscopy of CdIn_2_S_4_, Mo_2_C, and MS heterojunction composites ((**A**): full survey; (**B**): Cd 3d; (**C**): In 3d; (**D**): S 2p; (**E**): Mo 3d; (**F**): C 1s).

**Figure 6 molecules-28-02508-f006:**
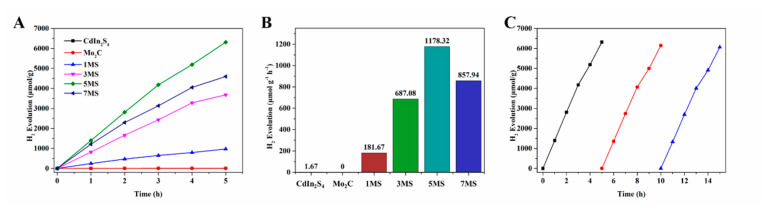
The amount of hydrogen produced over time (**A**); average rate of hydrogen production (**B**); stability of 5MS samples (**C**). Black is the first cycle, red is the second cycle, and blue is the third cycle.

**Figure 7 molecules-28-02508-f007:**
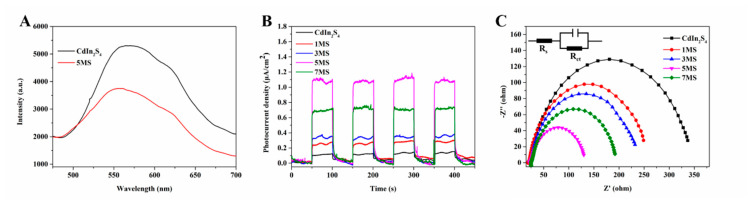
PL at an excitation wavelength of 469 nm (**A**); TPR (**B**) and EIS (**C**) for as-prepared samples.

**Figure 8 molecules-28-02508-f008:**
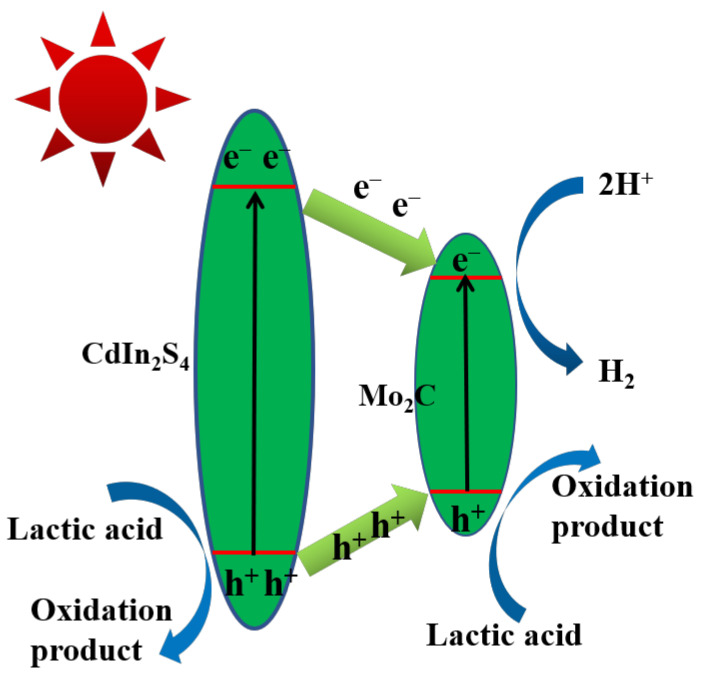
The photocatalytic hydrogen schematic diagram of MS heterojunction composites.

## Data Availability

New data have not been created.

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
