# Peer review of "Fabrication of Noble-Metal-Free Mo2C/CdIn2S4 Heterojunction Composites with Elevated Carrier Separation for Photocatalytic Hydrogen Production"

_molecules, 2023, doi:10.3390/molecules28062508_

Round 1
Reviewer 1 Report
The paper presented in this submission contains interesting results but their overall presentation and discussion lacks quality. A lot of characterization techniques are used to support the conclusions but comments on provided data are often very limited, too generic and not supported by proper references. Finally, the use of English language throughout the document is not up to standard. The grammar, sentence structure, and vocabulary choice impede the readability and clarity of the findings. It is recommended that the authors seek professional editing assistance to improve the language and enhance the overall impact of their work.
Please find here below more detailed comments.
1. Please correct chemical formula in the title and in the abstract (see capital letters and subscripts)
2. (Line 12) The heterojunction is conventionally defined “Type I” and not “I-type”. Please correct.
3. (Lines 24-28) The first eight references should be removed and replaced with more appropriate ones! All of these references are specific works published in 2022 on specific materials that clearly do not provide the general information set forth in this paragraph. Please replace all these references with much more relevant and appropriate ones such as DOI: 10.1016/j.mset.2019.12.002, DOI: 10.1016/j.rser.2014.10.101, DOI: 10.1016/j.rser.2021.111180.
4. (Line 97) How is the percentage estimated?
5. (Lines 122-124) Please provide a reference when mentioning “our previous reports”.
6. (Line 124) Please define which light source was used and its power density.
7. (Lines 135-136) The description of Figure 1A and 1B is too generic and wrong. Please provide an estimate of the size of these particles.
8. (Lines 145-148) Please add references to support the attribution of the lattice fringes to the specific planes of CdIn2S4 and Mo2C.
9. (Lines 157-158) Again, please add a reference describing this evidence.
10. (Line 166) Please replace “Fig. 8B” with “Fig. 3B”.
11. (Line 173) Please replace “Fig. 4C” with “Fig. 3C”.
12. (Figure 3A) The Y-axis should be labelled “Absorbance”.
13. (Lines 183-195) Please provide references for this data interpretation.
14. (Figure 6A and 6C) Please use “µmol g-1” as the unit of measure for the Y-axis to be consistent and clear for future comparison of this material with others reported in the literature.
15. (Figure 7C) A model circuit should be proposed to better describe the results as the discussion in the text is too generic.
Author Response
The paper presented in this submission contains interesting results but their overall presentation and discussion lacks quality. A lot of characterization techniques are used to support the conclusions but comments on provided data are often very limited, too generic and not supported by proper references. Finally, the use of English language throughout the document is not up to standard. The grammar, sentence structure, and vocabulary choice impede the readability and clarity of the findings. It is recommended that the authors seek professional editing assistance to improve the language and enhance the overall impact of their work. Please find here below more detailed comments.
- Please correct chemical formula in the title and in the abstract (see capital letters and subscripts)
RE:
First of all, thank you for your valuable suggestion. We appreciate for your warm reminder very much. We have corrected chemical formula in the title and in the abstract. Meanwhile, we carefully checked all the chemical formulas in the manuscript to make sure they were correct. We hope that the revised manuscript can meet the publishing requirements.
- (Line 12) The heterojunction is conventionally defined “Type I” and not “I-type”. Please correct.
RE:
Thank you for your correction. We have changed “I-type” to “Type I” in the revised manuscripts. We will pay more attention to this question in our subsequent work.
- (Lines 24-28) The first eight references should be removed and replaced with more appropriate ones! All of these references are specific works published in 2022 on specific materials that clearly do not provide the general information set forth in this paragraph. Please replace all these references with much more relevant and appropriate ones such as DOI: 10.1016/j.mset.2019.12.002, DOI: 10.1016/j.rser.2014.10.101, DOI: 10.1016/j.rser.2021.111180.
RE:
First of all, your suggestion is very important and valuable to our work. I have read the above literature and other reviews carefully. As suggested by the reviewer, all of the mentioned documents not only enrich our manuscript, but also provide the general information in this paragraph. Therefore, all papers have been cited as follows:
[1] A. Pareek, R. Dom, J. Gupta, J. Chandran, V. Adepu, P.H. Borse, Insights into renewable hydrogen energy: Recent advances and prospects, Materials Science for Energy Technologies, 3 (2020) 319-327.
[2] M. Wang, K. Han, S. Zhang, L. Sun, Integration of organometallic complexes with semiconductors and other nanomaterials for photocatalytic H2 production, Coordination Chemistry Reviews, 287 (2015) 1-14.
[3] X. Li, N. Li, Y. Gao, L. Ge, Design and applications of hollow-structured nanomaterials for photocatalytic H2 evolution and CO2 reduction, Chinese Journal of Catalysis, 43 (2022) 679-707.
[4] H. Ahmad, S.K. Kamarudin, L.J. Minggu, M. Kassim, Hydrogen from photo-catalytic water splitting process: A review, Renewable and Sustainable Energy Reviews, 43 (2015) 599-610.
[5] J. Ran, J. Zhang, J. Yu, M. Jaroniec, S.Z. Qiao, Earth-abundant cocatalysts for semiconductor-based photocatalytic water splitting, Chem Soc Rev, 43 (2014) 7787-7812.
[6] Y. Wang, Z. Ding, N. Arif, W.-C. Jiang, Y.-J. Zeng, 2D material based heterostructures for solar light driven photocatalytic H2 production, Materials Advances, 3 (2022) 3389-3417.
[7] M. Yue, H. Lambert, E. Pahon, R. Roche, S. Jemei, D. Hissel, Hydrogen energy systems: A critical review of technologies, applications, trends and challenges, Renewable and Sustainable Energy Reviews, 146 (2021) 111180.
[8] J. Corredor, M.J. Rivero, C.M. Rangel, F. Gloaguen, I. Ortiz, Comprehensive review and future perspectives on the photocatalytic hydrogen production, Journal of Chemical Technology & Biotechnology, 94 (2019) 3049-3063.
- (Line 97) How is the percentage estimated?
RE:
First of all, special thanks to you for considering to be published our manuscript. The questions and recommendations you proposed were very important to perfect our work. The percentage is the mass percentage, such as the heterojunction composites containing 0.2g of CdIn2S4 with 0.01g of Mo2C is labeled as 5MS photocatalysts. Similarly, the samples of CdIn2S4 with 1, 3 and 7 mass percent Mo2C are labeled as 1MS, 3MS and 7MS, respectively. We revised the description in the manuscript to make it clearer to the reader. We hope that the revised manuscript can meet the publishing requirements.
- (Lines 122-124) Please provide a reference when mentioning “our previous reports”.
RE:
We appreciate for your warm reminder very much. In order to make the reader understand, we provide a reference when mentioning “our previous reports” in the revised manuscript. We will pay more attention to this question in our subsequent work.
- (Line 124) Please define which light source was used and its power density.
RE:
First of all, your suggestion is very important and valuable to our work. We have provided relevant information in the manuscript considering the Xenon lamp used (CEL-HXF300F, 500W, China). Meanwhile, according to your suggestions, we measured the intensity of the incident radiation entering the photoreactor by an optical power meter (CEL-2000-2, China). Calculate the intensity of incident radiation by measuring the optical power density at nine different positions of the photoreactor according to the following formula:
E=PAverage*A
Where E, PAverage and A stand for the intensity of the incident radiation, the power density of light and the irradiation area, respectively.
The incident radiation intensity entering the photoreactor is shown in the table below.
|
1 |
2 |
3 |
4 |
5 |
6 |
7 |
8 |
9 |
Optical power density (mW/cm2) |
146.5 |
45.4 |
56.7 |
63.6 |
40.6 |
53.3 |
34.8 |
53.7 |
50.4
|
Average value (mW/cm2) |
60.6 |
||||||||
Reactor area (cm2) |
19.6 |
||||||||
Incident radiation intensity (mW) |
1186.9 |
- (Lines 135-136) The description of Figure 1A and 1B is too generic and wrong. Please provide an estimate of the size of these particles.
RE:
First of all, we sincerely appreciate you for reviewing our manuscript and we’re deeply grateful to your comment. In order to accurately describe the size of the photocatalyst, we modified the description of CdIn2S4 and Mo2C in the revised manuscript. Meanwhile, we will pay more attention to this question in our subsequent work.
- (Lines 145-148) Please add references to support the attribution of the lattice fringes to the specific planes of CdIn2S4 and Mo2C.
RE:
Thank you for your sincere suggestion. In order to support the attribution of the lattice fringes to the specific planes of CdIn2S4 and Mo2C (Lines 145-148), we provide a reference in the revised manuscript. We hope that the revised manuscript can meet the publishing requirements.
- Again, please add a reference describing this evidence.
RE:
We appreciate for your warm reminder very much. In order to support this description (Lines 157-158), we provide a reference in the revised manuscript. We will pay more attention to this question in our subsequent work.
- (Line 166) Please replace “Fig. 8B” with “Fig. 3B”.
RE:
Thank you for your correction. We have replaced “Fig. 8B” with “Fig. 3B” in the revised manuscripts. We will pay more attention to this question in our subsequent work.
- (Line 173) Please replace “Fig. 4C” with “Fig. 3C”.
RE:
Thank you for your correction. We have replaced “Fig. 4C” with “Fig. 3C” in the revised manuscripts. We will pay more attention to this question in our subsequent work.
- (Figure 3A) The Y-axis should be labelled “Absorbance”.
RE:
First of all, your suggestion is very important and valuable to our work. The Y-axis has changed as “Absorbance” in the revised manuscripts. We hope that the revised manuscript can meet the publishing requirements.
- (Lines 183-195) Please provide references for this data interpretation.
RE:
First of all, we sincerely appreciate you for reviewing our manuscript and we’re deeply grateful to your comment. To increase the credibility of XRD, we added some references to support the XRD results in the revised manuscripts. Moreover, we will pay more attention to this question you mentioned in our subsequent work.
- (Figure 6A and 6C) Please use “µmol g-1” as the unit of measure for the Y-axis to be consistent and clear for future comparison of this material with others reported in the literature.
RE:
Thank you for your sincere suggestion. The suggestion you proposed was very important to help our work. To be clear for future comparison of this material with others reported, we have changed unit of measure for the Y-axis as “µmol g-1” in the revised manuscripts. We hope that the revised manuscript can meet the publishing requirements.
- (Figure 7C) A model circuit should be proposed to better describe the results as the discussion in the text is too generic.
RE:
Thank you for your questions and well-meaning recommendation of our work. As you said, there is no model circuit in Fig. 7C. In order to better describe the results, we proposed a model circuit in Fig. 7C and modified the description of EIS in the revised manuscript. We will pay more attention to this question in our subsequent work.
Special thanks to you for your valuable comments and looking forward to learning more from you.
Reviewer 2 Report
Current manuscript describes fabrication of noble-metal-free Mo2C/CdIn2S4 heterojunction composites with elevated carrier separation for photocatalytic hydrogen production. I think it's so interesting and can be published after minor revision
1. The typo errors are observed in the title, e.g., "mo2c" "cdin2s4", as well as, in the whole of the manuscript. Please check it carefully.
2. Please rephrase the abstract. Abstract must be enriched with obtained data
3. Section 2.2: Did you use additional experiments for optimization of the amounts and conditions?. If yes, please clarify
4. Please compare the current work with similar work in this field.
Author Response
Current manuscript describes fabrication of noble-metal-free Mo2C/CdIn2S4 heterojunction composites with elevated carrier separation for photocatalytic hydrogen production. I think it's so interesting and can be published after minor revision
- The typo errors are observed in the title, e.g., "mo2c" "cdin2s4", as well as, in the whole of the manuscript. Please check it carefully.
RE:
First of all, thank you for your valuable suggestion. We appreciate for your warm reminder very much. We have corrected chemical formula in the title and in the abstract. Meanwhile, we carefully checked all the chemical formulas in the manuscript to make sure they were correct. Moreover, we will pay more attention to this question you mentioned in our subsequent work.
- Please rephrase the abstract. Abstract must be enriched with obtained data.
RE:
First of all, we sincerely appreciate you for reviewing our manuscript and we’re deeply grateful to your comment. We have added more obtained data to enrich the abstract. The abstract has been rephrased in the revised manuscript and we hope that the revised manuscript can meet the publishing requirements.
- Section 2.2: Did you use additional experiments for optimization of the amounts and conditions?. If yes, please clarify.
RE:
Thank you for your valuable recommendation. The question you raised had also aroused our attention and thinking. Optimization conditions of synthesis (e.g., temperature, time, etc.) are very important for photocatalyst to obtain superior performance. Based on the research of extensive literature and our previous research (Journal of Materials Chemistry A, 2021, 9, 14888-14896; Journal of Physics D: Applied Physics, 2020, 53 205101; CrystEngComm, 2021, 23, 5070-5077; Optical Materials, 2020, 108, 110231; Journal of Colloid and Interface Science, 2022, 628, 368-377; Journal of Colloid and Interface Science, 2023, 639, 87-95), we successfully synthesized pure CdIn2S4 at 180 ℃ for 24 h for better and convenient study. We have added related references into the revised manuscript. Meanwhile, we will pay more attention to the effects of reaction conditions on activity of photocatalysts and we will further explore it in subsequent research.
- Please compare the current work with similar work in this field.
RE:
First of all, special thanks to you for considering to be published our manuscript. The questions and recommendations you proposed were very important to perfect our work. According to your suggestion, we added the comparison of photocatalytic performance between this work and the current work with similar work in Supplementary Material Table S2. Compared with other current work, the Mo2C/CdIn2S4 photocatalyst may not have the greatest photocatalytic performance, but our work provides a possibility for the construction of heterojunction between Mo2C and sulfide photocatalyst to enhance H2 evolution activity. Meanwhile, we will try our best to improve CdIn2S4-based photocatalytic system in the future work.
Table S2. Comparison of hydrogen evolution data of Mo2C/CdIn2S4 composites compared with other literature reports.
Photocatalysts |
Noble metal |
H2 evolution |
Ref. |
Mo2C/CdIn2S4 |
No |
1178.32 μmol g-1 h-1 |
This work |
MoP/CdIn2S4 |
No |
286.10 μmol g-1 h-1 |
[1] |
Co2P/CdIn2S4 |
No |
471.87 μmol g-1 h-1 |
[2] |
Co9S8/CdIn2S4 |
No |
1083.6 μmol g-1 h-1 |
[3] |
CdIn2S4/CNFs/Co4S3 |
No |
25.87 mmol g-1 h-1 |
[4] |
ZnIn2S4/CdIn2S4 |
0.75 wt% PdS |
780 μmol h-1 |
[5] |
MoS2/CdIn2S4 |
No |
1868.19 μmol g-1 h-1 |
[6] |
MoSx/CdIn2S4 |
No |
2846.73 μmol g-1 h-1 |
[7] |
In2S3/CdIn2S4/In2O3 |
Pt |
2004 μmol g-1 h-1 |
[8] |
References
[1] X. Ma, W. Li, C. Ren, M. Dong, L. Geng, H. Fan, Y. Li, H. Qiu, T. Wang, Construction of novel noble-metal-free MoP/CdIn2S4 heterojunction photocatalysts: Effective carrier separation, accelerating dynamically H2 release and increased active sites for enhanced photocatalytic H2 evolution, J Colloid Interface Sci, 628 (2022) 368-377.
[2] X. Ma, W. Li, H. Li, M. Dong, L. Geng, T. Wang, H. Zhou, Y. Li, M. Li, Novel noble-metal-free Co2P/CdIn2S4 heterojunction photocatalysts for elevated photocatalytic H2 production: Light absorption, charge separation and active site, Journal of Colloid and Interface Science, 639 (2023) 87-95.
[3] C. Li, Y. Zhao, X. Liu, P. Huo, Y. Yan, L. Wang, G. Liao, C. Liu, Interface engineering of Co9S8/CdIn2S4 ohmic junction for efficient photocatalytic H2 evolution under visible light, J Colloid Interface Sci, 600 (2021) 794-803.
[4] S. Guo, Y. Li, C. Xue, Y. Sun, C. Wu, G. Shao, P. Zhang, Controllable construction of hierarchically CdIn2S4/CNFs/Co4S3 nanofiber networks towards photocatalytic hydrogen evolution, Chemical Engineering Journal, 419 (2021) 129213.
[5] Y. Yu, G. Chen, G. Wang, Z. Lv, Visible-light-driven ZnIn2S4/CdIn2S4 composite photocatalyst with enhanced performance for photocatalytic H2 evolution, International Journal of Hydrogen Energy, 38 (2013) 1278-1285.
[6] B. Zhang, H.X. Shi, X.Y. Hu, Y.S. Wang, E.Z. Liu, J. Fan, A novel S-scheme MoS2/CdIn2S4 flower-like heterojunctions with enhanced photocatalytic degradation and H2 evolution activity, Journal of Physics D: Applied Physics, 53 (2020) 205101.
[7] Q. Li, W.L. Liu, X.J. Xie, X.L. Yang, X.F. Chen, X.G. Xu, Synthesis and Characterization of Amorphous Molybdenum Sulfide (MoSx)/CdIn2S4 Composite Photocatalyst: Co-Catalyst Using in the Hydrogen Evolution Reaction, Catalysts 2020, 10, 1455.
[8] D. Ma, J.W. Shi, Y. Zou, Z. Fan, J. Shi, L. Cheng, D. Sun, Z. Wang, C. Niu, Multiple carrier-transfer pathways in a flower-like In2S3/CdIn2S4/In2O3 ternary heterostructure for enhanced photocatalytic hydrogen production, Nanoscale, 10 (2018) 7860-7870
Special thanks to you for your valuable comments and looking forward to learning more from you.
Reviewer 3 Report
molecules-2189652
Fabrication of noble-metal-free Mo2C/CdIn2S4 heterojunction composites with elevated carrier separation for photocatalytic hydrogen production
Authors
Hong Qiu , Xiaohui Ma , Hongxia Fan , Yueyan Fan , Yajie Li , Hualei Zhou * , Wenjun Li *
Section
The paper presents the synthesis of Mo2C/CdIn2S4 composites and claim that Mo2C/CdIn2S4 displayed high photocatalytic activity in visible range for H2 production (for 5Mo2C/CdIn2S4
1178.32 μmmol h-1 g-1).
The authors have achieved a well characterization of the composite, however the photocatalytic activity of Mo2C/CdIn2S4 composite requires a better scientific demonstration.
Rows 52-53: Why is Mo2C structurally similar with Pt?
"Mo2C has great potential for photocatalytic H2 evolution" Why?
Which is the band gap of Mo2C?
The reaction is the reduction of water to hydrogen.
Why the authors are using lactic acid for H2 production?
The band gap of pure CdIn2S4 is indirect with values between 2.1 eV
(corresponds to 590.40nm) and 2.4 eV (corresponds to 516.6 nm).
For the direct gap of CdIn2S4, the values are between 2.5
(corresponds to 495.93nm )
and 2.7 eV (corresponds to 516.6 nm) [Ref.Yohanna Seminovski, Pablo Palacios, Perla Wahnón, and Ricardo Grau-Crespo, Band gap control via tuning of inversion degree in CdIn2S4 spinel,
Appl. Phys. Lett. 100, 102112 (2012); https://doi.org/10.1063/1.3692780].
How the band gap is changed for the composite Mo2C/CdIn2S4 (or 5Mo2C/CdIn2S4)?
Why the photoluminescence spectra of CdIn2S4 and 5Mo2C/CdIn2S4
in the figure 7 are so similar? The red curve is just decreasing in intensity.
The values of the y axis (intensity) are missing.
The description of the PL is also missing.
Which excitation wavelength was used? Was it the whole emission of Xe lamp?
Author Response
The paper presents the synthesis of Mo2C/CdIn2S4 composites and claim that Mo2C/CdIn2S4 displayed high photocatalytic activity in visible range for H2 production (for 5Mo2C/CdIn2S4 1178.32 μmmol h-1 g-1). The authors have achieved a well characterization of the composite, however the photocatalytic activity of Mo2C/CdIn2S4 composite requires a better scientific demonstration.
- Rows 52-53: Why is Mo2C structurally similar with Pt?
RE:
First of all, special thanks to you for considering to be published our manuscript. The questions and recommendations you proposed were very important to perfect our work. In theoretical calculations and experiments, molybdenum carbide (Mo2C) has been proved to have an electronic structure similar to Pt. This result has been reported in previous studies (Angew. Chem., 2015, 127, 10902-10907; ACS Nano, 2014, 8 5164-5173; Energy Environ. Sci., 2013, 6, 1818-1826; J. Am. Chem. Soc. 2015, 137, 15753-15759; Renewable and Sustainable Energy Reviews, 2017, 75, 1101-1129; Nano Energy, 2018, 47, 463-473; Applied Surface Science, 2019, 470, 565-572; ChemSusChem, 2016, 9, 820-824). We've added a few more references to support this claim in the revised manuscript and hope that the revision will meet with approval.
- "Mo2C has great potential for photocatalytic H2 evolution" Why?
RE:
Thank you for your sincere questions. Molybdenum carbide (Mo2C), have been proven to show promise as low-cost alternatives to Pt catalysts in various catalytic processes dues to their similar electronic density of state to that of Pt and high electrical conductivity. In recent years, molybdenum carbide has attracted more and more attention in the field of photocatalysis. Not only has it been reported that it can be used for high-efficiency photocatalytic hydrogen production (J. Mater. Chem. A, 2017, 5, 10591-10598; Nano Energy, 2018, 47, 463-473; Applied Surface Science, 2019, 473, 91-101), but some studies have shown that catalytic performance of Mo2C is better than the precious metal (International journal of hydrogen energy, 2017, 42, 18977-18984; Journal of Alloys and Compounds, 2013, 569, 45-51; Journal of Colloid and Interface Science, 2021, 582, 488-495). Therefore, we mentioned in the introduction that molybdenum carbide has great potential.
- Which is the band gap of Mo2C?
RE:
First of all, we sincerely appreciate you for reviewing our manuscript and we’re deeply grateful to your comment. Based on our previous research and other literature (J Colloid Interface Sci, 582 (2021) 488-495; J. Mater. Chem. A, 5 (2017) 10591-10598; Nano Energy 47 (2018) 463-473), the band gap of Mo2C is severally 1.1 eV. We have highlighted the relevant descriptions in red in the revised manuscript and will pay more attention to this question in our subsequent work.
- The reaction is the reduction of water to hydrogen. Why the authors are using lactic acid for H2 production?
RE:
We’re so grateful for your important and valuable suggestions. As you said, at a pH of 7, water breaks down into H+ and OH-. After that, electrons in the conductor band of the photocatalyst will reduce the H+ to H2. Given that the reaction system is an aqueous solution of lactic acid (pH is about 2.2), the electrons will directly reduce H+ in the water to H2, and the holes will oxidize the lactic acid. We have added this information of pH in the revised manuscript. We hope that the revision will meet with approval.
- The band gap of pure CdIn2S4 is indirect with values between 2.1 eV (corresponds to 590.40 nm) and 2.4 eV (corresponds to 516.6 nm). For the direct gap of CdIn2S4, the values are between 2.5 (corresponds to 495.93 nm) and 2.7 eV (corresponds to 516.6 nm) [Ref.Yohanna Seminovski, Pablo Palacios, Perla Wahnón, and Ricardo Grau-Crespo, Band gap control via tuning of inversion degree in CdIn2S4 spinel, Appl. Phys. Lett. 100, 102112 (2012); https://doi.org/10.1063/1.3692780].
RE:
Thank you for your sincere comment. The question you raised had also aroused our attention and thinking. We have read the literature you mentioned carefully. At the same time, we also reviewed a lot of literature and found that different reports on the band gap of CdIn2S4 are different (2.0 eV - 2.4 eV) ( Nanoscale, 10 (2018) 7860-7870; Optical Materials 108 (2020) 110231; Journal of Colloid and Interface Science 600 (2021) 794-803; Chemical Engineering Journal 419 (2021) 129213; J. Mater. Chem. A, 9 (2021) 14888-14896; Catalysts, 10 (2020) 1455; Journal of Physics D: Applied Physics, 53 (2020) 205101). This may be caused by different synthesis conditions (e.g., temperature, time, raw materials, etc.) and different morphology of samples. In our manuscript, the band gap of CdIn2S4 was assessed as 2.3 eV, which is consistent with the results of most literature. Therefore, we think our results are reasonable and hope that you can recognize and agree with the revision.
- How the band gap is changed for the composite Mo2C/CdIn2S4 (or 5Mo2C/CdIn2S4)?
RE:
Thank you for your sincere suggestion. The suggestion you proposed was very important to help our work. In Fig. 3A, it was found that the absorption edge of heterojunction composites extended to longer wavelengths region compared with bare CdIn2S4, indicating that the band gap of the heterojunction composites is decreasing. According to the conversion, the band gap width of photocatalyst is obtained through DRS. The n values of different photocatalysts may be different. So, the band gap of heterojunction composites is only for reference. In Fig. S1, the band gap width of the heterojunction complex was calculated as 2.18 eV, 2.15 eV, 2.04 eV, 2.02 eV, respectively, which also verified the result that the band gap of heterojunction composites was decreasing. We will pay more attention to this question in our subsequent work and hope that the revision will meet with approval.
- Why the photoluminescence spectra of CdIn2S4 and 5Mo2C/CdIn2S4 in the Figure 7 are so similar? The red curve is just decreasing in intensity. The values of the y axis (intensity) are missing.
RE:
We’re so grateful for your important and valuable suggestions. In the manuscript, the emission spectrum of pure CdIn2S4 is at around 530 nm. Due to its intrinsic properties, different photocatalysts have different emission spectra. Therefore, pure Mo2C has no emission spectra at around 530 nm. Since 5MS composites contains CdIn2S4, the emission spectrum of 5MS composites is similar to that of CdIn2S4 in Fig. 7A. Generally, photoluminescence (PL) spectroscopy is one of the commonly used methods for evaluating the efficiency of photo-generated electron-hole pair charge separation. For photocatalysts, the lower PL emission peak intensity means the lower recombination of photogenerated charge. Fig. 7A display that MS composites possess lower peak than bare CdIn2S4, which means that composites have higher carrier separating efficiency after the CdIn2S4 is supported on the surface of Mo2C cocatalyst. The addition of Mo2C can increase the carrier separation of the catalyst, thus reducing the intensity of the red curve. As you said, the values of the y axis (intensity) are missing. We have changed Fig. 7A in the revised manuscript and will pay more attention to this question in our subsequent work.
- The description of the PL is also missing.
RE:
First of all, we sincerely appreciate you for reviewing our manuscript and we’re deeply grateful to your comment. We have been minutely described the results of the PL in the revised manuscript. We will pay more attention to this question in our subsequent work and hope that the revision will meet with approval.
- Which excitation wavelength was used? Was it the whole emission of Xe lamp?
RE:
We’re so grateful for your important and valuable suggestions. As you said, the Information about the excitation wavelength is missing in the manuscript. Xenon lamp is the light source for both the photocatalysis experiment and PL test. Photocatalysis experiment were carried out (λ ≥ 420 nm). And the excitation wavelength of PL is 280 nm. We have added the relevant information to the revised manuscript.
Special thanks to you for your valuable comments and looking forward to learning more from you.
Round 2
Reviewer 1 Report
The revised version of the manuscript can be accepted for publication.
Author Response
Special thanks to you for your valuable comments and looking forward to learning more from you.
Reviewer 3 Report
Pt is a catalyst (not a photocatalyst) for the hydrogen ion reduction. The principle of working of this is different from the principle of photocatalysis.
No band gap is involved once the Pt is metal not semiconductor.
Please correct these inaccuracies.
The electronic structure of Mo2C seems to be different from electronic structure of Pt if we analyse the number of electrons distributed on the orbitals:
Pt (Z=78) 1s2 2s22p6, 3s2 3p6, 4s23d104p6, 5s2 4d105p6, 6s2 4f14 5d8
which shifts to the more stable electronic structure:
1s2 2s22p6, 3s2 3p6, 4s23d104p6, 5s2 4d105p6, 6s1 4f14 5d9
The Pt has completed 4f orbitals (lantanide) being a 5d element, instead Mo2C contains 2 Mo, Mo is a 4d element with no 4f orbitals:
Mo (Z=42)
1s2 2s22p6, 3s2 3p6, 4s23d104p6, 5s2 4d4
which is more stable in the following electronic structure
1s2 2s22p6, 3s2 3p6, 4s23d104p6, 5s1 4d5
C (Z=6)
1s2 2s22p2
Mo2C has 90 electrons that need to organize themselves into an electronic structure. How can this be done similarly to Pt (pure metal with 78 electrons and a completed 4f orbitals)?
Without a reasoned explanation, the affirmation should be excluded from the article. Moreover Pt catalysis of H+ reduction to 0.5H2 is principially different from photocatalytic activities of Mo2C/CdIn2S4. In general, the photocatalysis creates the conditions for oxidation. How can be this used for reduction?
I mentioned before: The authors have achieved a well characterization of the composite, however the photocatalytic activity of Mo2C/CdIn2S4 composite requires a better scientific demonstration.
Author Response
Pt is a catalyst (not a photocatalyst) for the hydrogen ion reduction. The principle of working of this is different from the principle of photocatalysis. No band gap is involved once the Pt is metal not semiconductor.
Please correct these inaccuracies.
The electronic structure of Mo2C seems to be different from electronic structure of Pt if we analyse the number of electrons distributed on the orbitals:
Pt (Z=78) 1s2 2s22p6, 3s2 3p6, 4s23d104p6, 5s2 4d105p6, 6s2 4f14 5d8
which shifts to the more stable electronic structure:
1s2 2s22p6, 3s2 3p6, 4s23d104p6, 5s2 4d105p6, 6s1 4f14 5d9
The Pt has completed 4f orbitals (lantanide) being a 5d element, instead Mo2C contains 2 Mo, Mo is a 4d element with no 4f orbitals:
Mo (Z=42)
1s2 2s22p6, 3s2 3p6, 4s23d104p6, 5s2 4d4
which is more stable in the following electronic structure
1s2 2s22p6, 3s2 3p6, 4s23d104p6, 5s1 4d5
C (Z=6)
1s2 2s22p2
Mo2C has 90 electrons that need to organize themselves into an electronic structure. How can this be done similarly to Pt (pure metal with 78 electrons and a completed 4f orbitals)?
Without a reasoned explanation, the affirmation should be excluded from the article. Moreover Pt catalysis of H+ reduction to 0.5H2 is principially different from photocatalytic activities of Mo2C/CdIn2S4. In general, the photocatalysis creates the conditions for oxidation. How can be this used for reduction?
I mentioned before: The authors have achieved a well characterization of the composite, however the photocatalytic activity of Mo2C/CdIn2S4 composite requires a better scientific demonstration.
RE:
Thank you for your sincere suggestion. The suggestion you proposed was very important to help our work. According to your suggestion, we have deleted the sentence “Recently, Mo2C has gradually been studied for photocatalytic hydrogen production because of its Pt-like electronic structures”. Thank you for your recognition of the well characterization of our composites. Meanwhile, we also added some explanations based on the existing characterization to better support the photocatalytic activity results of Mo2C/CdIn2S4 composites. We will pay more attention to this question in our subsequent work and hope that the revised manuscript can meet the publishing requirements.
Special thanks to you for your valuable comments and looking forward to learning more from you.